# An Improved Method for Disassembly Depth Optimization of End-of-Life Smartphones Based on PSO-BP Neural Network Predictive Model

Shengqiang Jiao , Lin Li * , Fengfu Yin and Yang Yu

College of Electromechanical Engineering, Qingdao University of Science and Technology, Qingdao 266061, China; 4023031070@mails.qust.edu.cn (S.J.); Yinff@qust.edu.cn (F.Y.); 4023031045@mails.qust.edu.cn (Y.Y.)
* Correspondence: ll@qust.edu.cn; Tel.: +86-175-6173-7080

**Abstract**

Disassembly is a crucial step in the remanufacturing of end-of-life (EoL) electronic products. Disassembly depth refers to the disassembly stop point determined by the disassembly sequence. For the disassembly depth optimization of EoL electronic products, a feasibility model with a fast convergence and low mean squared error (MSE) is needed to improve optimization accuracy. However, the use of a backpropagation neural network (BPNN) model or mathematical model often results in a slow convergence and high MSE due to the randomness of the initial weights and biases. In this study, an improved method for the disassembly depth optimization of smartphones based on a Particle Swarm Optimization-BPNN (PSO-BPNN) predictive model is proposed. Compared with the traditional BPNN optimization method, the proposed method in this study is that the BPNN predictive model is optimized by using PSO, which shows a superior predictive performance and reduces the MSE. The case of 'Huawei P7' is used to verify the feasibility of the method. The results show that the method maintains disassembly profit while reducing the disassembly time and carbon emissions by 17.1% and 7.8%, respectively. Compared with the BPNN model, the PSO-BPNN model converges 18.6%, 32.8%, and 16.6% faster in predicting the disassembly time, profit, and carbon emissions, respectively, with MSE reductions of 92.95%, 96.51%, and 92.74%, respectively.

**Keywords:** disassembly depth optimization; EoL smartphones; PSO-BPNN; MSE; pareto solutions



## 1. Introduction

Nowadays, the rapid development of science and technology has accelerated product updates, which results in the generation of many EoL products [1]. Waste electrical and electronic equipment (WEEE) is one of the fastest-growing waste streams globally, with EoL smartphones accounting for a significant proportion [2]. For example, in China, about 30 million EoL smartphones are generated annually, but the recycling rate is even less than 1%, with most being discarded or incinerated [3,4]. With the increasing amount of WEEE, EoL smartphones are an important component that cannot be ignored due to their variety and high recycling value [5]. In order to recycle resources and protect the environment, many countries have carried out stricter legislation and have paid more attention to the recycling and remanufacturing of EoL products, such as the "Product Recycling Law" in Europe and the "Recyclability Law" in Japan [6,7].

Disassembly depth optimization is a process aimed at finding the optimal disassembly sequence that can meet the set optimization objectives, and the disassembly sequence may not be a complete disassembly of EoL smartphones. As a crucial step in the recycling process of EoL products, disassembly has an irreplaceable role in sustainable development [8]. Compared to complete disassembly, partial disassembly is more practical and has been widely adopted, becoming the mainstream choice for many disassembly enterprises [9,10]. Yin et al. [11] proposed a partial sequence-dependent disassembly line balancing problem considering tool requirements. To avoid unnecessary work and labor waste, Yin et al. [12] proposed a multimanned partial disassembly line balancing problem. Determining the optimal disassembly depth for improving efficiency and reducing costs has always been a concern for enterprises and researchers [13]. Wu et al. [14] solved the EoL Li-ion recycling problem by extending the disassembly depth of Li-ion batteries. Gao et al. [15] constructed a partially observable multi-intelligence reinforcement learning environment that combines the structure of an electric vehicle battery with a disassembly task for disassembly depth optimization. Hu et al. [16] proposed a generic ontology model and combined it with a rule-based reasoning method to automatically generate the optimal disassembly depth. Furthermore, for the research on the disassembly depth optimization of EoL electronic products, the feasibility of model plays a significant role in determining the final research results. Therefore, it is necessary to provide a feasibility optimization model for disassembly depth optimization [17].

Currently, there are two types of optimization models within the literature: one is mathematical models based on intrinsic mechanisms and the other is neural network models with multiple inputs and multiple outputs [18]. Mathematical optimization methods have been applied in manufacturing industries [19]. The mathematical models include the disassembly time model [20], disassembly difficulty model [21], and disassembly profit model [22]. Xing et al. [23] established a disassembly time model considering execution time and preparation time to solve the asynchronous parallel disassembly sequence planning problem. Yang et al. [24] considered efficiency, profit, and environmental impact to establish a disassembly time, $CO_2$ emission, and recycling cost model aimed at solving the disassembly line optimization problem for EoL products. Liu et al. [25] proposed a disassembly time and profit model to improve the success rate of the disassembly operation in disassembly sequence planning. Tuo et al. [26] introduced a multi-channel disassembly line balancing model considering pedestrian workers. Although the above researchers have already solved most of the problems related to disassembly optimization by establishing mathematical models, in the face of multi-objective nonlinear or cross-domain systems, mathematical models have difficulty capturing their behaviors. Moreover, assumptions are often required to deal with complex nonlinear problems, which lead to a decrease in the accuracy and feasibility of the mathematical models [27].

Compared to mathematical models, machine learning can be effectively used to generate models to avoid the above limitations [28]. Common neural network models include an identify model based on Bayesian regularization optimization [29], a non-parametric model based on an improved adaptive neural network [30], and a Eulerian constitutive model for rate-dependent inelasticity enhanced by a neural network [31]. In addition, neural networks were used in various research areas because of their nonlinear learning, adaptive, and generalization abilities [29]. The neural network model has the advantages of a high identification accuracy and low cost. However, if only a single neural network model is used to solve the problem, the convergence speed is slow and the dependence on data is strong [32], which will lead to an infeasible model.

According to previous studies, neural networks have developed into primary optimization models for solving related optimization problems. In fact, with the rapid advancement of deep learning technology, neural networks have gradually evolved from a simple function approximation tool into a general optimization framework [33]. At present, most researchers combine optimization algorithms with neural network models to solve the problem of model unfeasibility. Wang et al. [34] optimized the BPNN model using the Sparrow Search Algorithm; its training error and predictive error are less than 5%, and its $R^2$ value is close to 1. Chen et al. [35] introduced the LSTM model and the BPNN model improved by the SSA algorithm to predict high-complexity and low-complexity subsequences, respectively; the results show that the accuracy of the improved model can reach 99.17%. Through a hyperparameter search, Wang et al. [36] constructed a 14-layer deep BP neural network model; an SGD algorithm with a Mini-batch algorithm was adopted to effectively balance the model performance and the training time complexity, resulting in a 76.68% performance improvement and a nearly 6-times training time reduction. Wu et al. [37] proposed a combined IFOA-BP neural network. The IFOA-BP model and IFOA cooperate to find the best combination of design parameters; compared with a contrastive design, the obtained optimization set of design parameters can reduce the self-weights by about 6.85%. Overall, these studies have demonstrated that optimized neural network models exhibit superior performance. However, few researchers have applied the optimized neural network model to the disassembly depth optimization research of EoL electronic products. Table 1 presents a comparison of the classical literature in the field of disassembly model establishment research in recent years.

**Table 1.** Comparison of the classical literature in the field of disassembly model establishment.

| Reference | Product Type | DM | | | OM | | Objectives | | | | | |
|---|---|---|---|---|---|---|---|---|---|---|---|---|
| | | CD | PD | MM | NNM | INNM | DS | DE | DT | DP | DC | SI |
| [38] | Automobile engine | + | | + | | | | | + | | + | |
| [39] | Used hard disk drive | + | | + | | | + | | + | | | |
| [40] | Gear pumps | + | | + | | | | + | | | + | |
| [13] | EoL smartphone | | + | + | | | + | + | + | + | | |
| [41] | Refrigerator disassembly line | + | | + | | | | | | + | | + |
| [42] | Refrigerator disassembly | + | | + | | | | | + | + | + | |
| [43] | EoL battery packs | + | | | + | | + | | + | | | |
| [44] | Retired power battery | + | | | | + | | | | + | + | |
| [45] | NEV-P50 battery | + | | | + | | + | | + | | | |
| [46] | Gear pumps | + | + | + | | | | | | + | | |
| [24] | Corn harvester cutting table | + | | + | | | | + | + | | + | |
| [47] | U-shaped disassembly lines | + | | + | | | + | | + | | | + |
| This study | EoL smartphone | | + | | | + | | | + | + | + | |

DM: disassembly model; CD: complete disassembly; PD: partial disassembly; OM: optimization model; MM: mathematical model; NNM: neural network model; INNM: improved neural network model; DS: disassembly safety; DE: disassembly energy consumption; DT: disassembly time; DP: disassembly profit; DC: disassembly carbon emissions; SI: smoothness index.

To further illustrate the differences in various machine learning models and deep learning models, Table 2 lists the features of different predictive models based on the parameters/hyperparameters of the models. All works mentioned above have made great achievements in the field of disassembly model establishment. Most researchers choose to take the traditional mathematical model as the optimization model. However, it is difficult to fully capture their behavior in nonlinear problems for using mathematical models. In addition, the use of the BPNN model on its own, due to the randomness of initial weights and biases, will cause gradient vanishing or explosion and reduce the convergence speed

and predictive performance, which will also affect the optimization process. To address those drawbacks, this study proposes an improved method for the disassembly depth optimization of EoL electronic products based on a PSO-BPNN predictive model. The PSO-BPNN predictive model is capable of rapidly fitting multiple performance metrics for the disassembly depth of EoL electric products, while effectively seeking global optimized pareto solutions through intelligent optimization. This method is particularly well-suited for scenarios where multiple nonlinear objectives are mutually restrictive, making it a preferred method in current disassembly depth optimization research.

**Table 2.** The characteristics of different predictive models in machine learning/deep learning.

| Predictive Model | Parameters/Hyperparameters | Applicable Scenarios |
| --- | --- | --- |
| RF | Tree structure and integrated parameters | Classification/regression tasks |
| SVM | Kernel function and regularization parameters | Small sample classification and high-dimensional space problems |
| LSTM | Loop structure and training parameters | Time series data prediction |
| GBM | Integrate and iterate parameters | Structured data regression/classification |
| MLP | Network structure and optimization parameters | Nonlinear regression/classification |
| LR | Regularization and optimization parameters | Binary classification task |
| DT | Tree structure parameters | Simple classification/regression |
| BPNN | Network structure and training parameters | Nonlinear fitting, pattern recognition |

The contribution of the present work was twofold. Firstly, this study introduced a data-driven modeling method, which eliminated reliance on traditional mathematical models. Traditional methods usually rely on simplified assumptions, which can reduce the accuracy of the model for dealing with the complex nonlinear characteristics of the disassembly process. By using neural networks, the complex relations between the disassembly depth optimization process and the optimization results were learned. This eliminated a series of assumptions and provided a more feasible optimization model. Secondly, this study has constructed a systematic methodological framework that covered the entire process of prediction and optimization and provided a technical solution for addressing the problem of disassembly depth optimization for EoL smartphones. This framework integrated data-driven prediction models with multi-objective optimization algorithms, which achieved the entire process from generating a feasible disassembly depth to the acquisition of the optimal disassembly depth. The method proposed in this study provided a feasible test for researching the disassembly depth optimization field of EoL electronic products and offered a reference for subsequent research.

## 2. Disassembly Depth Optimization Method Based on PSO-BPNN Model

The method proposed in this study can be divided into five main steps, as shown in Figure 1. In the first step, the disassembly parts are numbered, a disassembly directed graph is constructed based on actual disassembly experiments, the disassembly constraint matrix and rules are defined, and feasible disassembly depths are generated. The second step is to optimize the initial weights and biases of the BPNN predictive model by using

the PSO. In the third step, the experimental data is randomly divided into training and test sets at a rate of 7:3; the number of training iterations is 1000, the learning rate is 0.01, and the target minimum error is $1 \times 10^{-5}$. Based on the experimental results, the PSO-BPNN model was established to predict disassembly time, profit, and carbon emissions. In the fourth step, the PSO-BPNN predictive model is embedded into the Non-dominated Sorting Genetic Algorithm (NSGA-II) and uses its predicted outputs as fitness values. In the fifth step, NSGA-II is used for the disassembly depth optimization of EoL smartphones to balance disassembly time, profit, and carbon emissions.

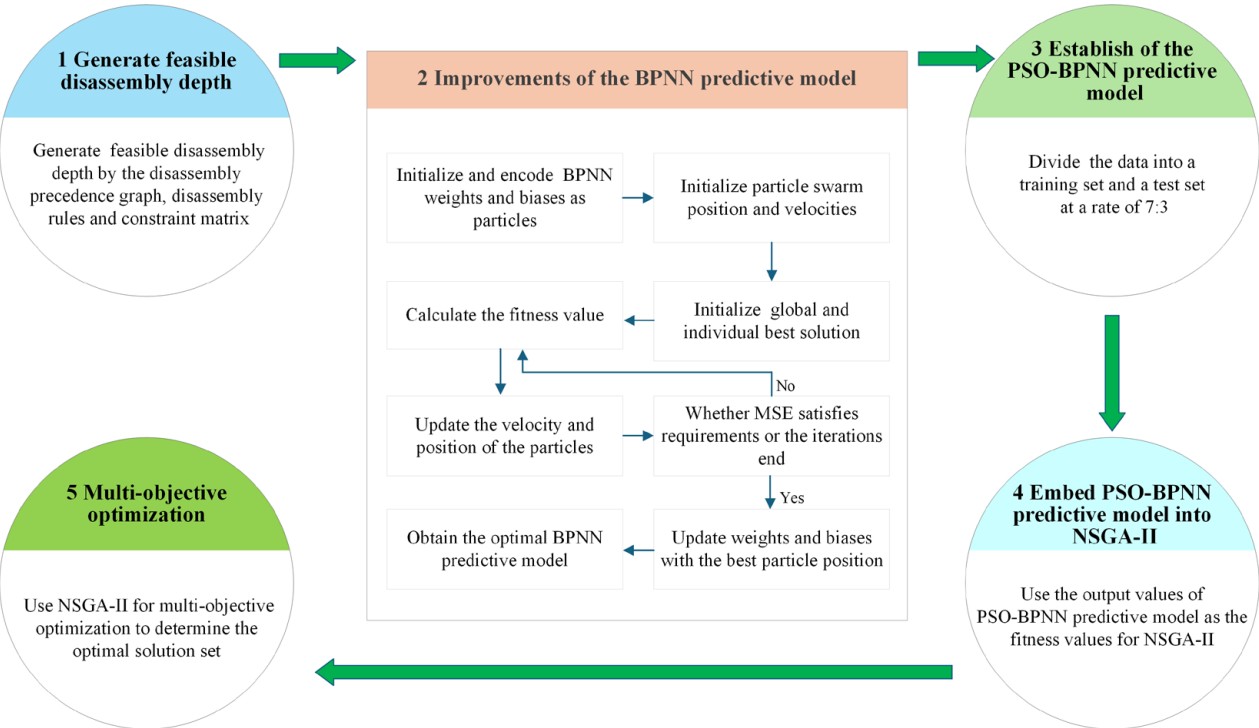

**Figure 1.** The flow chart of the disassembly depth optimization method.

### 2.1. Generate Feasible Disassembly Depth

The disassembly precedence graph is commonly used to represent the relationships between parts of the EoL product [14], as shown in Figure 2. The disassembly precedence graph is constructed by a human expert based on a physical teardown analysis, and the precedence graph truly reflects the constraints in the actual disassembly. A single-direction arrow represents constraint relationships between parts—for example, an arrow from part 1 to part 2 means part 1 must be disassembled before part 2. A solid arc represents an 'AND' priority relationship, which means all preceding disassembly tasks need to be completed before subsequent disassembly can begin. A dashed arc represents an 'OR' priority relationship, where subsequent disassembly tasks can proceed as soon as any one preceding task is finished. The disassembly precedence graph can be represented by the constraint matrix $P$.

$$\left[ P_{i,j} \right]_{11 \times 11} = \begin{pmatrix} P_{1,1} & \cdots & P_{1,11} \\ \vdots & \ddots & \vdots \\ P_{11,1} & \cdots & P_{11,11} \end{pmatrix} \tag{1}$$

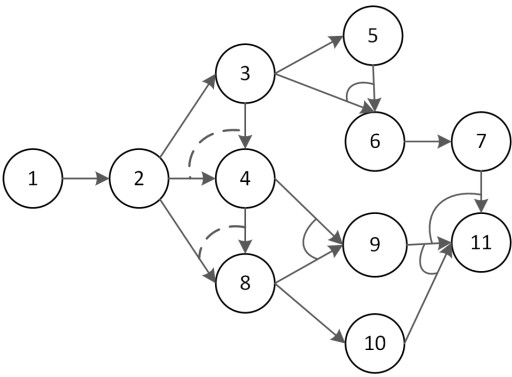

**Figure 2.** The disassembly precedence graph.

Matrix *P* is used to describe the types of tasks; for example, $P_{1,3} = 0$ indicates that part 3 cannot be disassembled after removing part 1, $P_{1,2} = 1$ means part 2 can be disassembled once part 1 is removed, and $P_{2,4} = 2$ shows that part 4 can be selectively disassembled after removing part 2.

The disassembly rules for EoL smartphones were established as follows [13]: (1) the disassembly of parts with a high recycling value; (2) the disassembly of parts with a high environmental pollution impact; (3) the disassembly of parts that require the same disassembly tool; and (4) the disassembly of parts with the same attributes.

### 2.2. Improvements of the BPNN Predictive Model

The BPNN is a typical multi-layer feedforward artificial neural network trained using the error backpropagation algorithm. The core principle of BPNN involves a two-phase process: a forward pass, where input signals are processed layer-by-layer to generate an output, and a back pass, where the error between the actual output and the target is calculated and propagated backward through the network. Through this iterative process, the connection weights and biases of each neuron are adjusted to minimize the error. The BPNN model mainly consists of the input layer, hidden layers, and output layer [27]. Random initial weights and biases in the BPNN predictive model often results in a slow convergence speed, high MSE, and prediction inaccuracies. Therefore, this study uses PSO to optimize the weights and biases of the BPNN predictive model. Figure 3 illustrates the topological structure graph of the PSO-BPNN model.

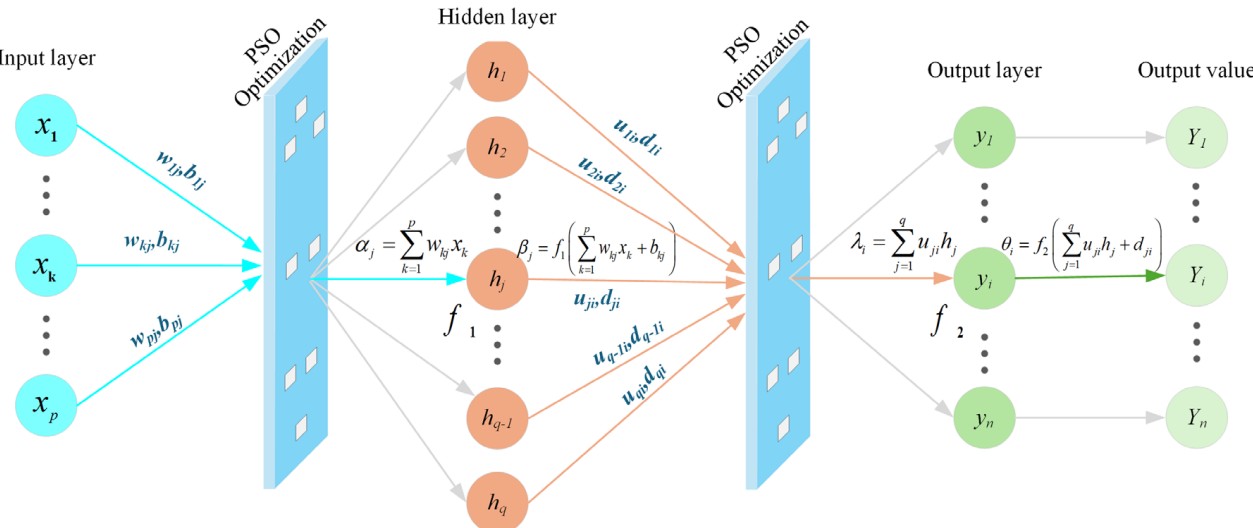

**Figure 3.** PSO-BPNN predictive model topological structure graph.

From Figure 3, the computational principle of BPNN can be observed [30]. The network processes $P$ independent input variables through an initial weighting operation using the weight matrix $W = [w_{11}, w_{12}, \ldots, w_{jk}, \ldots, w_{pq}]$ and bias vector $B = [b_{11}, b_{12}, \ldots, b_{kj}, \ldots, b_{pq}]$. These weighted inputs then pass through hidden layer neurons, where activation function $f_1$ transforms the signals. Similarly, the output layer subsequently performs another weighted summation using weight matrix $U = [u_{11}, u_{12}, \ldots, u_{ji}, \ldots, u_{qs}]$ and bias vector $D = [d_{11}, d_{12}, \ldots, d_{ji}, \ldots, d_{qs}]$, followed by a transformation through activation function $f_2$. Compared to other functions, the smooth and differentiable nature of Sigmoid makes it more compatible with gradient-based optimizers like PSO, leading to a more efficient and stable parameter search. For both layers, the Sigmoid function is employed as the nonlinear activation operator.

$$f_{1,2}(x) = Sigmoid(x) = \frac{1}{1 + e^{-x}} \tag{2}$$

The input and output of the $j$-th hidden layer neuron and the $i$-th output layer neuron can be calculated using Equations (3)–(6).

$$\alpha_j = \sum_{k=1}^{p} w_{kj} x_k \tag{3}$$

$$\beta_j = f_1 \left( \sum_{k=1}^{p} w_{kj} x_k + b_{kj} \right) \tag{4}$$

$$\lambda_i = \sum_{j-1}^{q} u_{ji} h_j \tag{5}$$

$$\theta_i = f_2 \left( \sum_{j=1}^{q} u_{ji} h_j + d_{ji} \right) \tag{6}$$

where $p$ represents the number of input layer neurons, $q$ represents the number of hidden layer neurons, and $x_k$ is the value of the $k$-th input layer neuron ($k = 1, \ldots, p$). The connection between input and hidden layers is characterized by weights $w_{jk}$ and biases $b_{jk}$, which links the $k$-th input layer neuron to the $j$-th hidden layer neuron. Similarly, $h_j$ is the value of the $j$-th input hidden layer neuron ($j = 1, \ldots, q$) and $u_{ij}$ and $d_{ji}$ are the weights and biases from the $j$-th hidden layer neuron to the $i$-th output layer neuron, respectively. The weight matrices can be represented by Equations (7) and (8).

$$q_1 = \begin{pmatrix} w_{11} & \cdots & w_{1q} \\ \vdots & \ddots & \vdots \\ w_{p1} & \cdots & w_{pq} \end{pmatrix} \tag{7}$$

$$q_2 = \begin{pmatrix} u_{11} & \cdots & u_{1n} \\ \vdots & \ddots & \vdots \\ u_{q1} & \cdots & u_{qn} \end{pmatrix} \tag{8}$$

where $q_1$ is the weight matrix from the input layer to the hidden layer and $q_2$ is the weight matrix from the hidden layer to the output layer. In addition, the number of neurons in the input and output layers of BPNN is determined by variable dimensions, and the number of hidden layer neurons is usually determined based on empirical Formula (9).

$$s = \sqrt{n + l} + a \tag{9}$$

where $a$ is a constant ranging from [1, 10], $l$ is the number of neurons in the output layer, and $n$ is the number of neurons in the input layer.

After model training, the predictive performance of the PSO-BPNN model is typically evaluated using MSE [48]. MSE quantifies the average squared difference between the predicted values and the actual values of the model, which serve as a key indicator for assessing prediction accuracy. A lower MSE indicates a better model performance and a higher prediction accuracy. In addition, the $R^2$ is used to assess the goodness of fit, with higher values indicating that the model explains a greater proportion of the variance in the data.

$$MSE = \frac{1}{n}\sum_{i=1}^{n}(Y_i - y_i)^2 \tag{10}$$

$$R^2 = 1 - \frac{\sum_{i=1}^{n}(Y_i - y_i)^2}{\sum_{i=1}^{n}(Y_i - \bar{y})^2} \tag{11}$$

where $Y_i$ is the actual value, $y_i$ is the predicted value from the model, $\bar{y}$ is the average of the actual value, and $n$ is the number of samples.

The MSE is used as the objective function for PSO, which iteratively optimizes the weights and biases of BPNN. The optimization procedure consists of the following steps.

Step 1: The weights and biases of the BPNN are encoded as particles. The positions xi and velocities vi of the particle swarm are initialized, with each particle representing a BPNN predictive model. The position represents a combination of weights and biases.

Step 2: Initialize the global best solution ($p_g$) and the individual best solution ($p_i$), which represent the optimal solution of the particle swarm and each individual particle, respectively.

Step 3: Use the MSE of the BPNN as the fitness function to calculate the fitness value of each particle.

Step 4: Based on the individual best solution and the global best solution of particles, combined with the inertia weights ($w$), learning factors ($c_1$, $c_2$), and random numbers ($r_1$, $r_2$), the new velocity and position are calculated by the following equation [49].

$$\begin{cases} v_i^{k+1} = \omega v_i^k + c_1 r_1(p_i - x_i^k) + c_2 r_2(p_g - x_i^k) \\ x_i^{k+1} = x_i^k + v_i^{k+1} \end{cases} \tag{12}$$

where $v_i^k$ and $x_i^k$ represent the velocity and position (combination of weights and biases) of particle $i$ in the $k$-th generation, respectively.

Step 5: Update the global best solution and individual best solution based on the fitness values. If the fitness value of a particle is better than the global best solution, update the global best solution; otherwise, update the individual best solution.

Step 6: If the MSE satisfies the defined requirements or the iterations end, output the current best position as the optimal weights and biases of the BPNN predictive model; otherwise, return to Step 4 and continue the iteration.

Step 7: The optimal weights and biases are assigned to the BPNN predictive model. If the MSE satisfies the defined requirements, output the predictive value; otherwise, return to Step 1.

### 2.3. Establishment of the PSO-BPNN Predictive Model

The PSO is used in the traditional BPNN predictive model to optimize the weights and biases of the model. Firstly, the network structure needs to be determined, which includes the number of neurons in the input layer, hidden layer, and output layer. Secondly, the particle swarm is initialized, where each particle represents the weights and biases of the BPNN predictive model, with randomly generated initial values to ensure diversity. During

training, MSE is used as an evaluation index to measure the performance of the network on the training data, which served as the objective function of PSO. Finally, through iterative optimization, the network tended to the optimal weights and biases, and the optimal weights and biases were assigned to the BPNN predictive model. It should be noted that, although the initial estimate of disassembly carbon emission is based on disassembly time and carbon emissions factors, the carbon emissions of each disassembly step may not be proportional to the disassembly. Therefore, the final model predicts it as an independent nonlinear function of multiple operations.

The number of input layer neurons corresponds to the relevant feature variables in the problem domain. For the disassembly depth optimization of EoL smartphones, the input layer features include the basic disassembly time, disassembly cost, part recovery price, part recovery rate, indirect carbon emissions, and sorting time. Therefore, the number of input layer neurons $x_p$ is set to six. The number of output layer neurons depends on the predictive targets. In this study, the output layer is defined to include the disassembly time, profit, and carbon emissions. Therefore, the number of output layer neurons $y_n$ is set to three. The number of hidden layer neurons $h_q$ is determined to be six based on Equation (9). The 150 experimental data samples are randomly divided into training and testing sets at a ratio of 7:3, which aim to better train the model. The neural network employs a Sigmoid activation function across all layers, with hidden layer training performed through the Levenberg–Marquardt algorithm. The training process is configured with a maximum of 1000 iterations and a learning rate of 0.01, and the target minimum error is $1 \times 10^{-5}$ to ensure optimal model performance while preventing overfitting.

Accordingly, a 6-6-3 BPNN predictive model is established. The established BPNN predictive model adjusts the weights and biases of each neuron through feedback from the error function, which enables it to produce outputs similar to the expected target distribution.

### 2.4. Embed the PSO-BPNN Predictive Model into NSGA-II

In terms of predicting the performance of the optimization objectives, compared with the traditional BPNN predictive model, the PSO-BPNN predictive model demonstrates a strong learning and generalization ability, and the accuracy of the prediction of disassembly time, profit, and carbon emissions has also been improved. As shown in Figure 4, the trained PSO-BPNN predictive model generated predictive outputs (disassembly time, profit, and carbon emissions) for each disassembly depth, which serve as the objective functions for the subsequent NSGA-II optimization process.

NSGA-II introduces the elite retention strategy based on the original NSGA to prevent the loss of excellent individuals during the evolution process and improve the convergence of the algorithm [50]. In addition, NSGA-II does not require the pre-allocation of weights for each objective function, which avoids the subjectivity of weight selection and makes it applicable to a wider range of problems. The main steps for the PSO-BPNN predictive model to be embedded into NAGA-II are as follows.

Step 1: Establish the predictive model for disassembly time, profit, and carbon emissions, respectively.

Step 2: Train the PSO-BPNN predictive model until iteration ends or MSE satisfies the defined requirements.

Step 3: The trained PSO-BPNN predictive model generates output predicted results: disassembly time, profit, and carbon emissions.

Step 4: NSGA-II randomly generates the initial population, with the first generation selected randomly within the variable range.

Step 5: The PSO-BPNN prediction results serve as the fitness values for individuals in the optimization process.

Step 6: Calculate the crowding distance of each individual to preserve solution diversity along the pareto front. Individuals with larger crowding distances are given a higher selection priority, as they represent less densely populated regions of the objective space. The calculation formula is as follows.

$$d_i = \sum_{m-1}^{M} \left( \frac{f_m(i+1) - f_m(i-1)}{f_m^{\max} - f_m^{\min}} \right) \tag{13}$$

where $d_i$ is the crowding distance of individual $i$ and $f_m(i-1)$ and $f_m(i+1)$ are the objective values of the neighboring individuals, where $f_m^{\max}$ and $f_m^{\min}$ are the maximum and minimum values of the objective, respectively.

Step 7: Select individuals based on fitness values and crowding distance, with priority given to those with higher fitness and smaller crowding distances for the next generation.

Step 8: Perform crossover and mutation to generate better solutions. If the crossover occurs at positions $k_1$ and $k_2$, the formula is as follows.

$$\begin{cases} O_1 = \left[ x_1^1, \ldots, x_{k_1}^1, x_{k_1+1}^2, \ldots, x_{k_2}^2, x_{k_2+1}^1, \ldots, x_n^1 \right] \\ O_2 = \left[ x_1^2, \ldots, x_{k_1}^2, x_{k_1+1}^1, \ldots, x_{k_2}^1, x_{k_2+1}^2, \ldots, x_n^2 \right] \end{cases} \tag{14}$$

where $O_1$ and $O_2$ represent the parent individuals.

Step 9: To obtain the pareto solutions, repeat Steps 6 to 8 until the iteration ends.

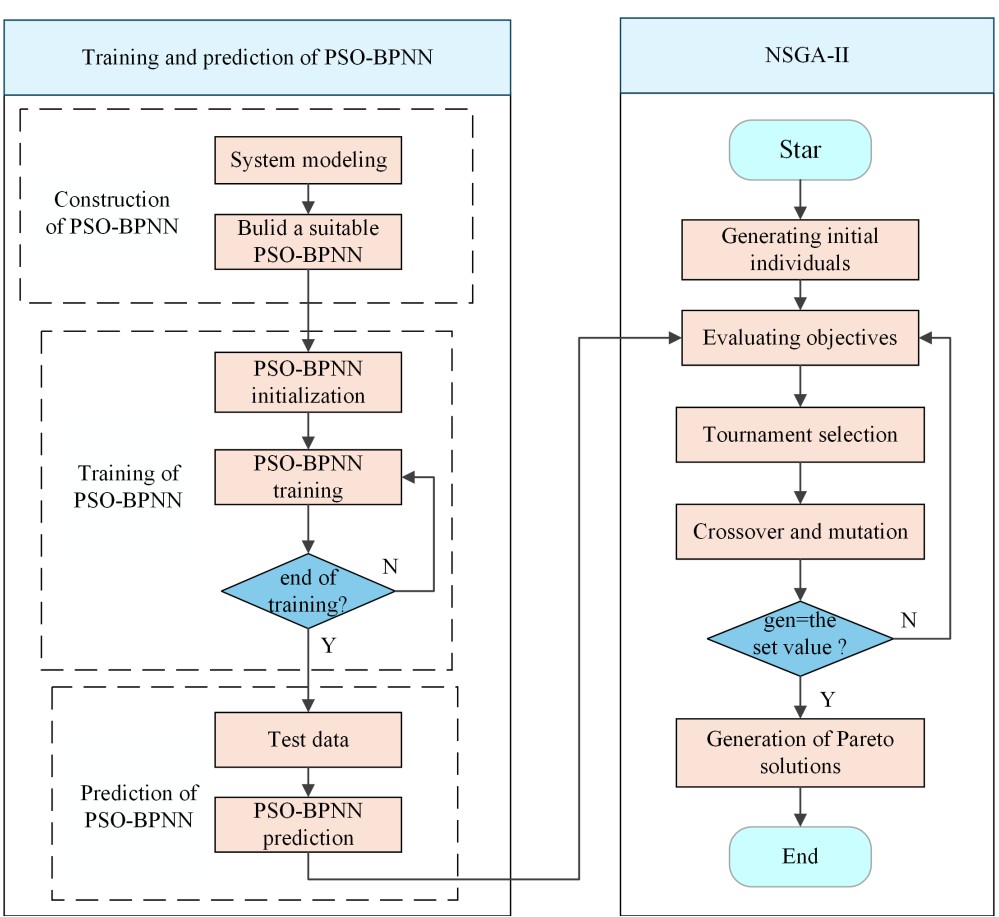

**Figure 4.** The algorithm flow chart of embedding PSO-BPNN into NSGA-II.

*2.5. Multi-Objective Optimization*

In consideration of the practical circumstances of relevant disassembly companies, the disassembly time, profit, and carbon emissions were used as optimization objectives in this study. There are conflicts between optimization objectives. To present the optimal solutions obtained more clearly, the NSGA-II is used to seek the pareto front. Disassembly time is calculated by the weighted sum of the times for each stage, disassembly profit is based on the value of recovered parts, recovery rates, and costs, and disassembly carbon emissions are estimated based on the energy consumption. To achieve a better convergence on the pareto front, an appropriate number of iterations and population size are necessary [51]. The initial population size is set to 20 to ensure sufficient diversity. The iterations of NSGA-II are set to 100 to enable the algorithm to search effectively and converge on suitable solutions. The crossover and mutation probabilities are set to 0.9 and 0.1, respectively.

Commonly, the maximum disassembly profit is the desired outcome. However, the core theory of the optimization algorithm is mainly based on the gradient descent principle—a method that conducts iterative searches by following the steepest decent path to find the optimal solution. Therefore, in multi-objective optimization problems, objectives are typically minimized. The pareto optimization problem is formulated as follows.

$$\min f(x) = \min\left[T, \frac{1}{P}, E\right] \tag{15}$$

where $T$, $P$, and $E$ represent the disassembly time, disassembly profit, and disassembly carbon emissions, respectively. $T \in [0, 446]$, $P \in [-20.5, 65.5]$, $E \in [169.5, 337.5]$. Since our research aims to maximize the disassembly profit while minimizing the disassembly time and disassembly carbon emissions, the objective function for $T$ and $E$ can be directly formulated for minimization. To align with this minimization framework, a minimization of the reciprocal of the objective function $P$ is performed.

## 3. Case Study

The EoL smartphone, namely 'Huawei P7', is used as an example to verify the feasibility of the proposed method. The PSO-BPNN and BPNN predictive models are used to predict disassembly time, profit, and carbon emissions, respectively. Combined with NSGA-II, the predictive output of the model is optimized, and the feasibility of the proposed method is verified.

*3.1. Fundamental Data*

The disassembly part graph for 'Huawei P7' is shown in Figure 5, which contains the 12 main parts of 'Huawei P7' and their numbers. The precedence graph of 'Huawei P7' shows the precedence constraint relationship between the 12 main parts, as shown in Figure 6. In addition, the data in this study are derived from actual disassembly experiments and market research. Due to the relative comparison being used, the research results are not affected by part recovery prices. The disassembly tool information is shown in Table 3, and the tool damage cost is estimated using the wholesale purchase prices and service life statistics [13]. The basic disassembly information of 'Huawei P7' is shown in Table 4, which includes part market recovery rates, recovery prices, and other relevant parameters. Table 5 lists the relevant carbon emission factors, which depend on reference [52] and mainly consider the fuel consumption of material transportation and the electricity consumption for equipment operation.

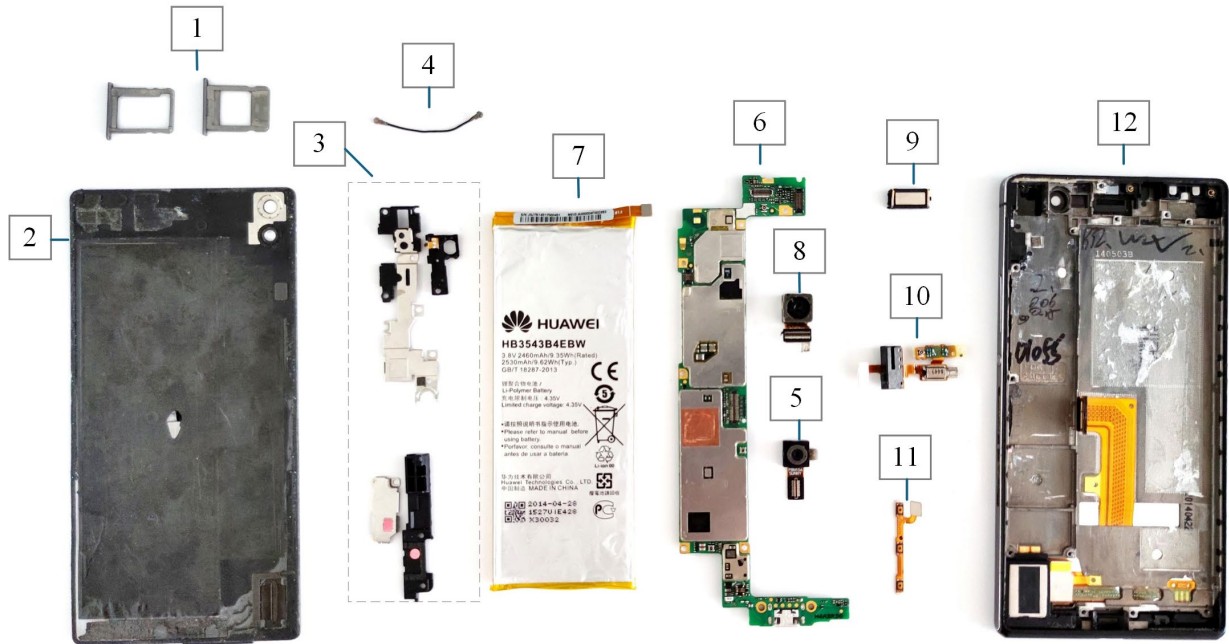

**Figure 5.** Disassembly part graph of 'Huawei P7'.

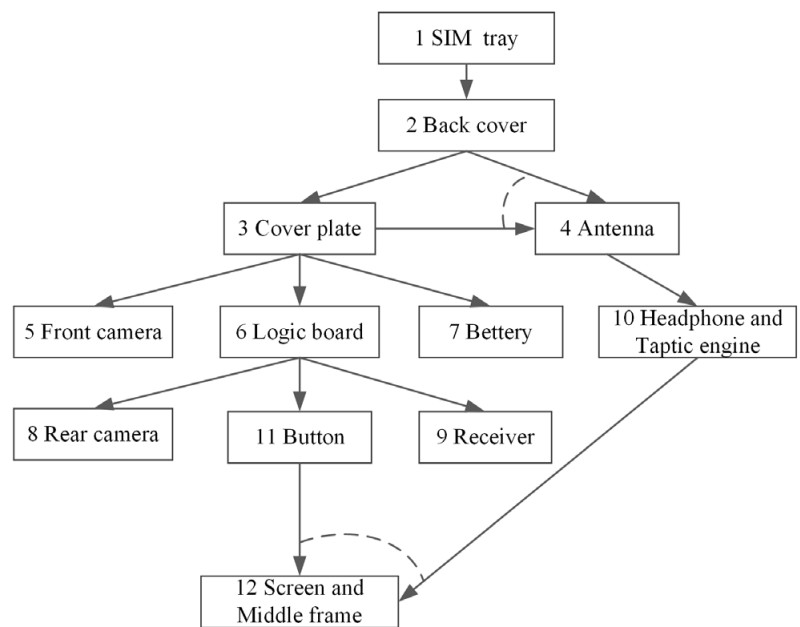

**Figure 6.** The disassembly precedence graph of 'Huawei P7'.

**Table 3.** Disassembly tool information.

| Tool Name | Tool Operational Parameters | The Tool Damage Cost for Each Use |
|---|---|---|
| Electric screw bit | Power: 50 w; Rotational speed: 500 r/min | 0.01 |
| Hot air gun | Power: 600 w; Heating temperature: 100 °C | 0.01 |
| SIM-ejector tool | Moving speed: 0.5 m/s | 0.005 |
| ESD-safe tweezer | Moving speed: 0.5 m/s | 0.005 |
| Black stick | Moving speed: 0.5 m/s | 0.005 |

**Table 4.** The basic disassembly information of 'Huawei P7'.

| Part No. | Part Name | Connection Method of Parts | Disassembly Tools and Usage Sequence | Disassembly Direction | Market Recovery Rate | Market Recovery Price (CNY) |
|---|---|---|---|---|---|---|
| 1 | SIM tray | Insert | SIM-ejector tool | −Y | 95% | 0.5 |
| 2 | Back cover | Adhesive | Hot air gun → Black stick | +Z | 95% | 0.5 |
| 3 | Cover plate | Screws + Buckle | Electric screw bit | +Z | 95% | 5 |
| 4 | Antenna | Snap-fit | Black stick | +Z | 95% | 1 |
| 5 | Front camera | Clip-on Connector | Black stick | +Z | 98% | 15 |
| 6 | Logic board | Screws + Snap-fit | Electric screw bit → Black stick | +Z | 99% | 45 |
| 7 | Battery | Adhesive | Hot air gun → Black stick | +Z | 99% | 20 |
| 8 | Rear camera | Clip-on Connector | Black stick | +Z | 98% | 20 |
| 9 | Receiver | Adhesive | ESD-safe tweezer | +Z | 95% | 0.5 |
| 10 | Headphone and Taptic engine | Adhesive + Snap-fit | Black stick | +Z | 95% | 0.5 |
| 11 | Button | Adhesive + Snap-fit | ESD-safe tweezer | +Z | 95% | 0.5 |
| 12 | Screen and Middle frame | Adhesive | Hot air gun → Black stick → ESD-safe tweezer | −Z | 98% | 1.5 |

**Table 5.** Carbon emissions factor of different sources.

| Energy Type | Utilization (*M*) | Carbon Emission Factors (*C*) |
|---|---|---|
| Electrical energy | Power consumption of equipment ($M_1$) | $C_1 = 0.5$ kg/kw h |
| Diesel fuel | Power consumption of material transportation ($M_2$) | $C_2 = 2.31$ kg/L |
| Gasoline | Power consumption of material transportation ($M_3$) | $C_3 = 3.68$ kg/L |

The disassembly time is measured using a stopwatch, and the disassembly carbon emissions ($E_{co2}$) and disassembly cost (*N*) are calculated by Equations (15) and (16).

$$E_{co_2} = M \times C \tag{16}$$

$$N = \sum_{l=1}^{q} (u_l \times g_l) + N_2 + N_3 + N_4 \tag{17}$$

where *M* represents the energy consumption, *C* is the carbon emission factor, *q* is the number of disassembly tools used, $u_l$ is the number of times each smartphone uses tool *l*, and $g_l$ is the tool damage cost of using tool *l* once. $N_2$ and $N_3$ represent labor costs and the wholesale price of smartphones, respectively, while $N_4$ indicates the price of energy consumption.

The dataset used in this study was constructed based on the actual disassembly experiment data. To ensure consistency, all disassembly processes followed the standardized procedure derived from the disassembly precedence graph. For each operation in the process, the disassembly was precisely recorded. The final dataset consists of 150 independent records, covering the features and results throughout the complete disassembly process.

### 3.2. Disassembly Schemes

Based on the basic disassembly information of 'Huawei P7', the disassembly precedence graph, and the disassembly rules, the feasible disassembly depth is generated. As shown in Figure 7, to avoid experimental randomness, three personnel with experience in EoL smartphone disassembly were selected to conduct the experiment simultaneously, and the disassembly data were recorded, respectively. In the data processing procedure, to reduce experimental costs and generate irregular samples, Latin Hypercube Sampling (LHS) was used to randomly select 50 sets of disassembly depths [53]. The LHS design is formulated as shown in Equation (17).

$$x_j^{(i)} = \frac{\pi_j^{(i)} + U_j^{(i)}}{n_S} \quad , 1 \leq j \leq n_s, \quad 1 \leq i \leq n_s \tag{18}$$

where $i$ is the $i$-th sample, $j$ is the $j$-th component of $j$-th sample, $U_j$ is taken from the range [0, 1], $n_s$ is the number of sample points, and its value is a random number from [0, n−1].

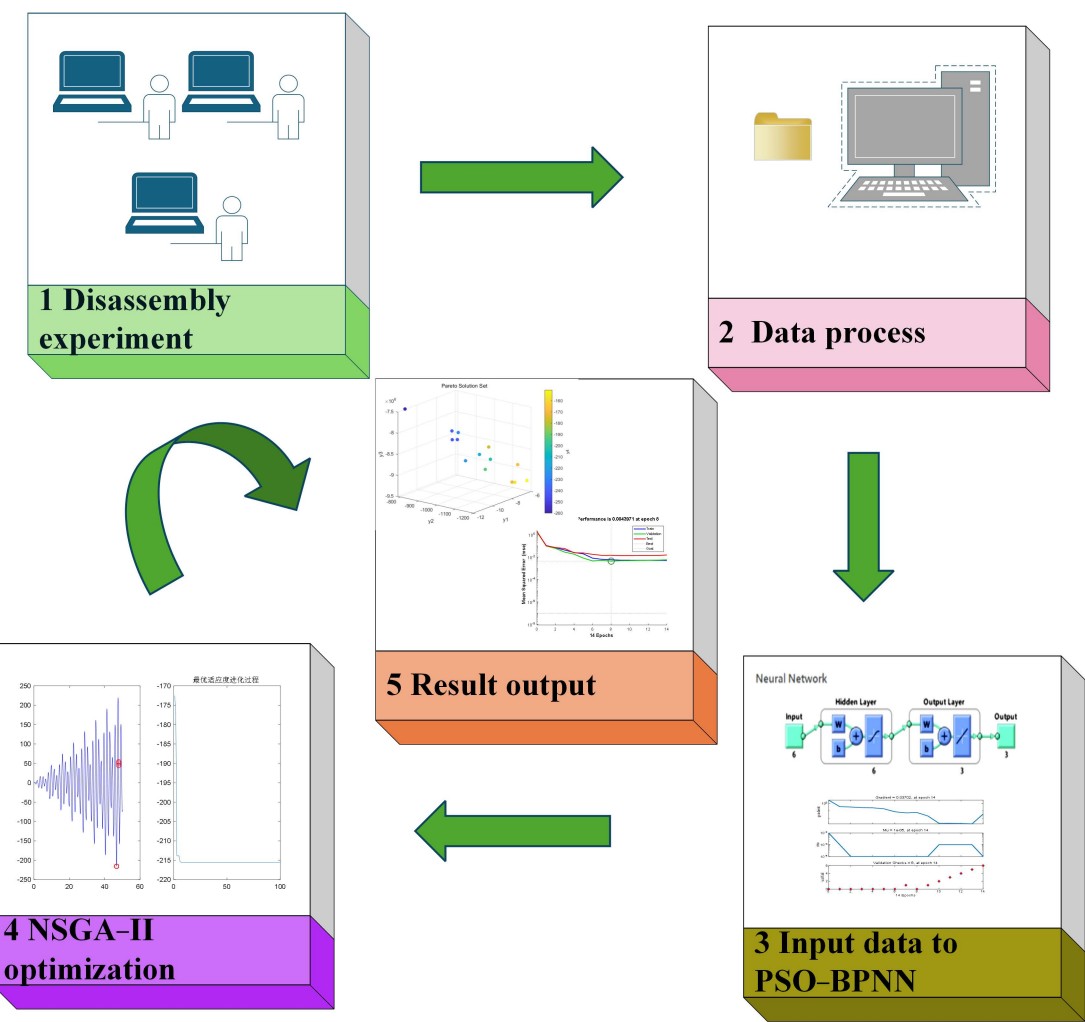

**Figure 7.** The application flow chart of proposed method in the case study.

The basic disassembly time, disassembly cost, part recovery price, part recovery rate, indirect carbon emissions, and sorting time are all important factors; therefore, they are taken as the input for the neural network. Then, the processed data is input into the PSO-BPNN predictive model, and the predicted results are used as the individual fitness values for NSGA-II optimization. Finally, the pareto solutions were obtained.

## 4. Results and Discussion

Firstly, this section discusses the predictive performance of the PSO-BPNN predictive model for the disassembly time, profit, and carbon emissions of EoL smartphones. The comparative analysis demonstrates that the superior performance of the PSO-BPNN predictive model is better than the BPNN predictive model. Secondly, for the disassembly depth optimization of EoL smartphones, the superior pareto solutions are obtained by the proposed method.

### 4.1. Discussion for Prediction Results of the PSO-BPNN Predictive Model

Three groups of experiments were conducted to verify the predictive performance of the PSO-BPNN predictive model. Furthermore, to provide a baseline reference, the multiple linear regression (MLR) predictive model was also compared with the BPNN predictive model and PSO-BPNN predictive model.

In the first experimental group, 150 sets of data were selected, and the training set and the test set were divided into a ratio of 7:3 for verification. The input and output samples were trained by the PSO-BPNN and BPNN predictive models to establish the network relationships between inputs and corresponding outputs. Figure 8 shows the disassembly time prediction results for the three models on the test set. As shown in Figure 8a, the MLR predictive model exhibits a poor predictive accuracy. In Figure 8b, the BPNN predictive model exhibits a significant prediction bias for samples 43 to 45. In contrast, Figure 8c shows that the PSO-BPNN predictive model maintains a strong agreement between the predictive values and actual values across all samples. According to Equation (10), the MSE values for the MLR, BPNN, and PSO-BPNN predictive models are calculated as 66.523, 58.022, and 4.083, respectively. Compared to the BPNN predictive model, the PSO-BPNN predictive model achieves an approximately 92.95% reduction in MSE. Therefore, the predictive model established based on PSO-BPNN was effective in the first group experiment.

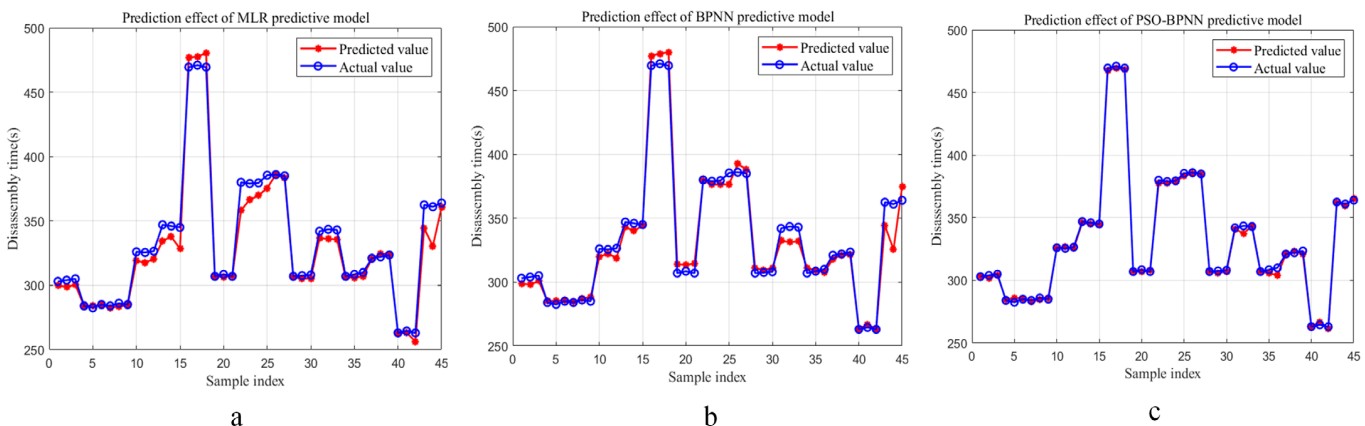

**Figure 8.** Disassembly time prediction results ((**a**) MLR; (**b**) BPNN; (**c**) PSO-BPNN).

The same procedure was applied to the second and third sample groups, and similar results were obtained. The MLR predictive model also has a poor prediction accuracy in the second and third sample groups. Compared with the BPNN predictive model, the PSO-BPNN predictive model still maintains a relatively high accuracy, which further verified the predictive performance of the PSO-BPNN predictive model for both disassembly profit and carbon emissions. Figures 9 and 10 show the results of the second and third experiments, respectively. The quantitative analysis shows that, for disassembly profit prediction, the MSE values are 12.422 (BPNN) and 0.433 (PSO-BPNN), which represents an approximately 96.51% reduction. Similarly, for carbon emission prediction, the corresponding MSE values are 13.178 (BPNN) and 0.956 (PSO-BPNN), which demonstrates an approximately 92.74%

reduction. From Figure 11, the $R^2$ of the PSO-BPNN predictive model reaches 0.9964, 0.9946, and 0.9936 for disassembly time, disassembly profit, and disassembly carbon emissions, respectively, higher than that of BPNN predictive model (0.9779, 0.9713, 0.9785). Although both models have been fully trained, the PSO-BPNN predictive model has an excellent predictive ability for this study case. In addition, all prediction accuracy is also higher than the trusted value ($R^2$ = 0.9092) in the literature [49]. Therefore, the PSO-BPNN predictive model could be accepted as the optimization model for subsequent disassembly depth optimization.

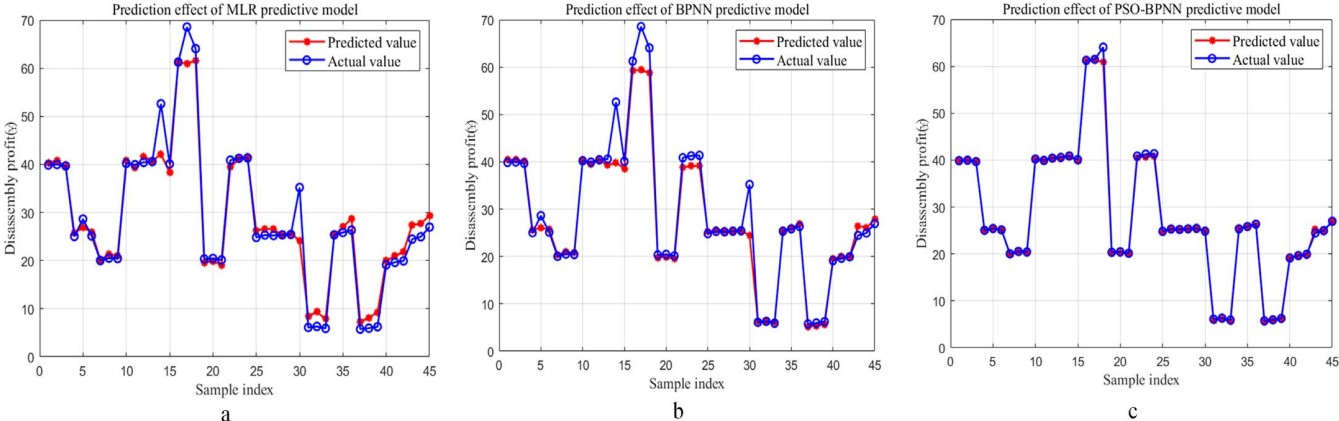

**Figure 9.** Disassembly profit prediction results ((**a**) MLR; (**b**) BPNN; (**c**) PSO-BPNN).

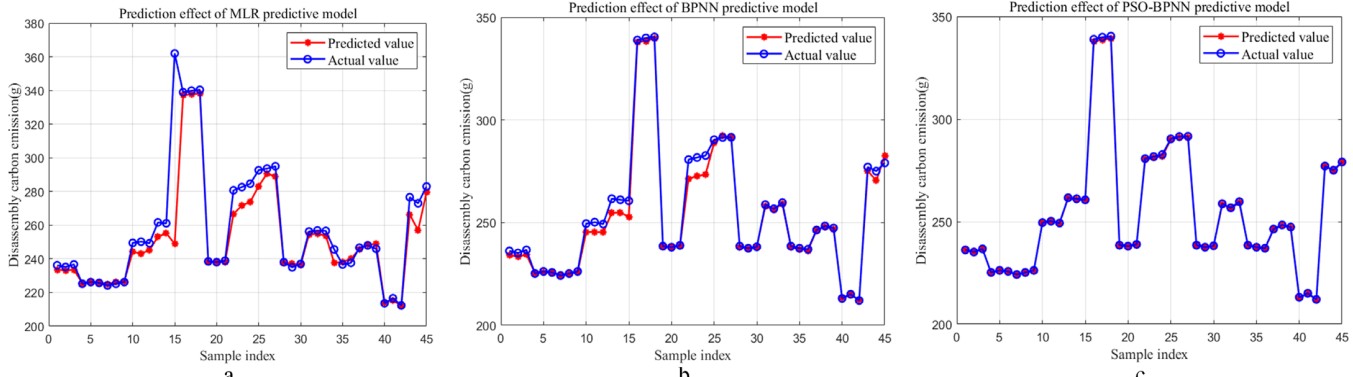

**Figure 10.** Disassembly carbon emissions prediction results ((**a**) MLR; (**b**) BPNN; (**c**) PSO-BPNN).

The accuracy of the PSO-BPNN predictive model for disassembly time, disassembly profit, and disassembly carbon emissions does not reach 100% for some reasons. Firstly, as a data-driven approximate model, the predictive ability of PSO-BPNN is limited by the network structure and optimization algorithms. If the number of network layers is too small or the number of neurons is insufficient, the model may not be able to fully learn complex data patterns, and the network may not converge. Secondly, the small number of samples may result in poor model training with regard to the predictive ability. In addition, the learning rate curve of the proposed model is shown in Figure 12. In the initial 0–3 epochs, both the training and validation errors decreased rapidly, indicating that the model is effectively learning the features of the data. Between epochs 3 and 10, the error rate slows down as the model converged, with the validation error achieving the optimal performance. From epoch 10 to 6, the training error continues to decrease slightly, but the validation and test errors begin to increase marginally, indicating that the model continues to fit the training set. The training error has been reduced to a very low level (at an order of $10^{-3}$) and is significantly lower than in the initial stage. Therefore, the model does

not suffer from severe underfitting. After the 10th epoch, the training error continues to decrease, while the validation and test errors stabilized or slightly increased, which are signs of overfitting.

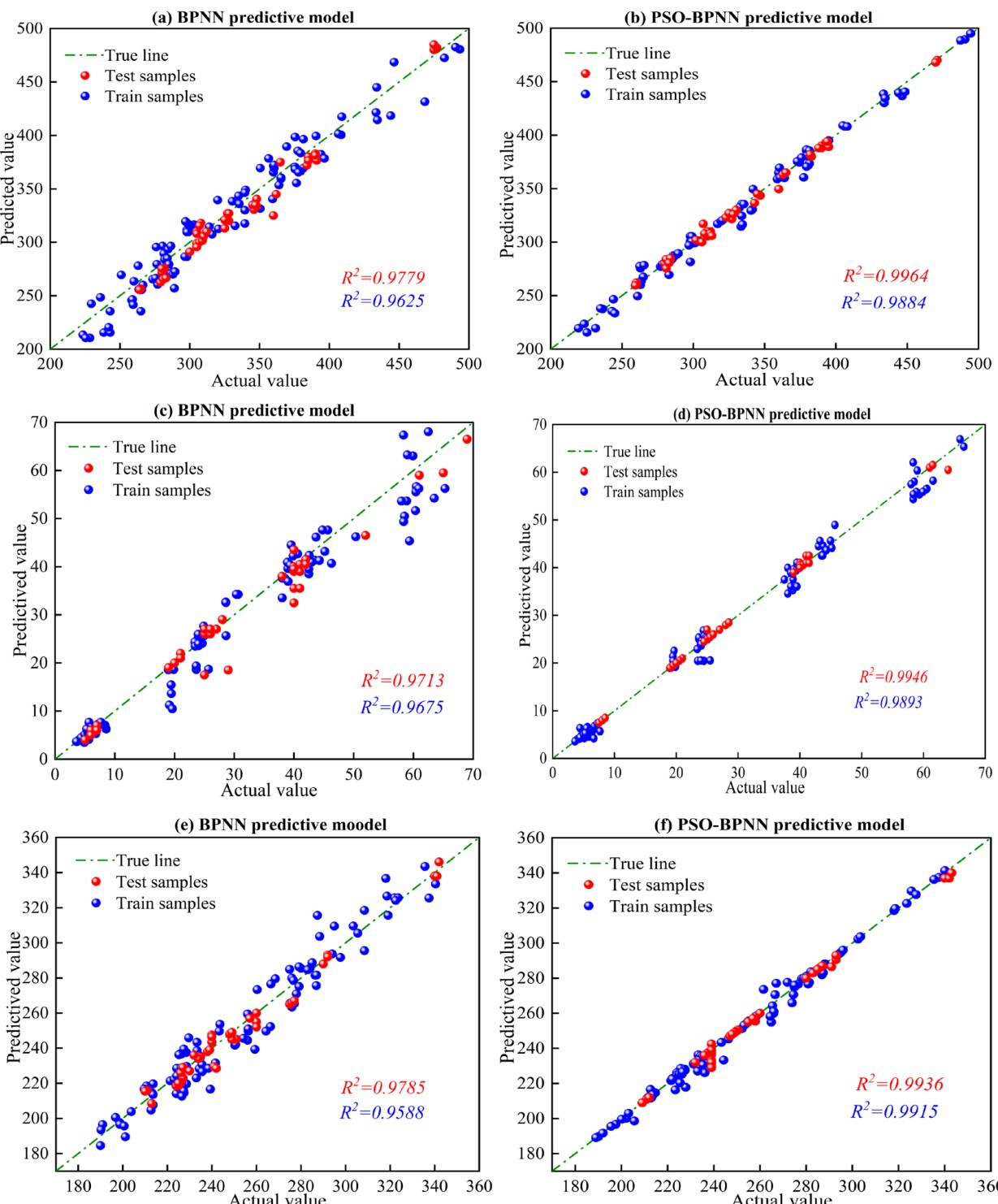

**Figure 11.** Comparison in regression between BPNN and PSO-BPNN. (**a**) Regression of BPNN for disassembly time; (**b**) regression of PSO-BPNN for disassembly time; (**c**) regression of BPNN for disassembly profit; (**d**) regression of PSO-BPNN for disassembly profit; (**e**) regression of BPNN for disassembly carbon emissions; (**f**) regression of PSO-BPNN for disassembly carbon emissions.

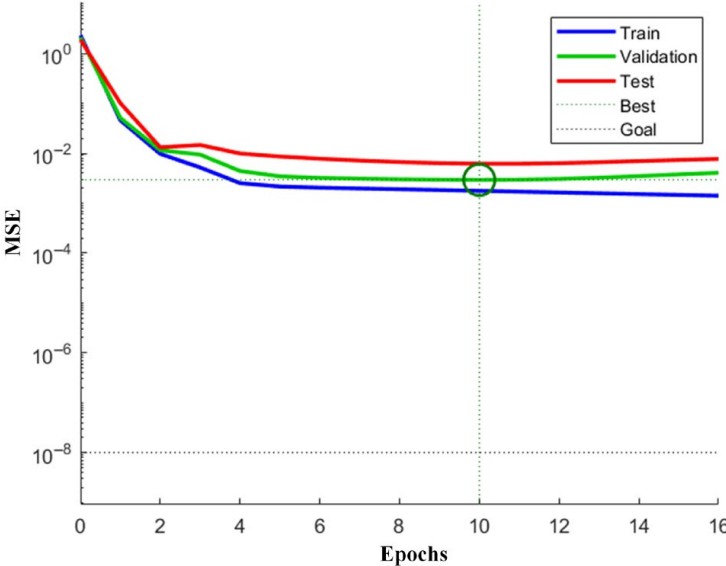

**Figure 12.** The performance graph of the neural network training process.

### 4.2. Discussion for the Pareto Optimal Solutions

This section presents and compares the disassembly depth optimization results obtained from both the BPNN optimization method and the proposed method in this study. Figure 13 presents the pareto solutions obtained by both methods. Specifically, Figure 13a shows those obtained by the traditional BPNN optimization method, while Figure 13b shows the pareto solution obtained by the proposed method in this study. The circular markers in the figure represent different disassembly depths. For easy analysis, the pareto solutions were divided into two groups: group A (traditional BPNN optimization method) and group B (proposed method in this study). Table 6 shows the results of the disassembly depth optimization for the EoL smartphone 'Huawei P7'.

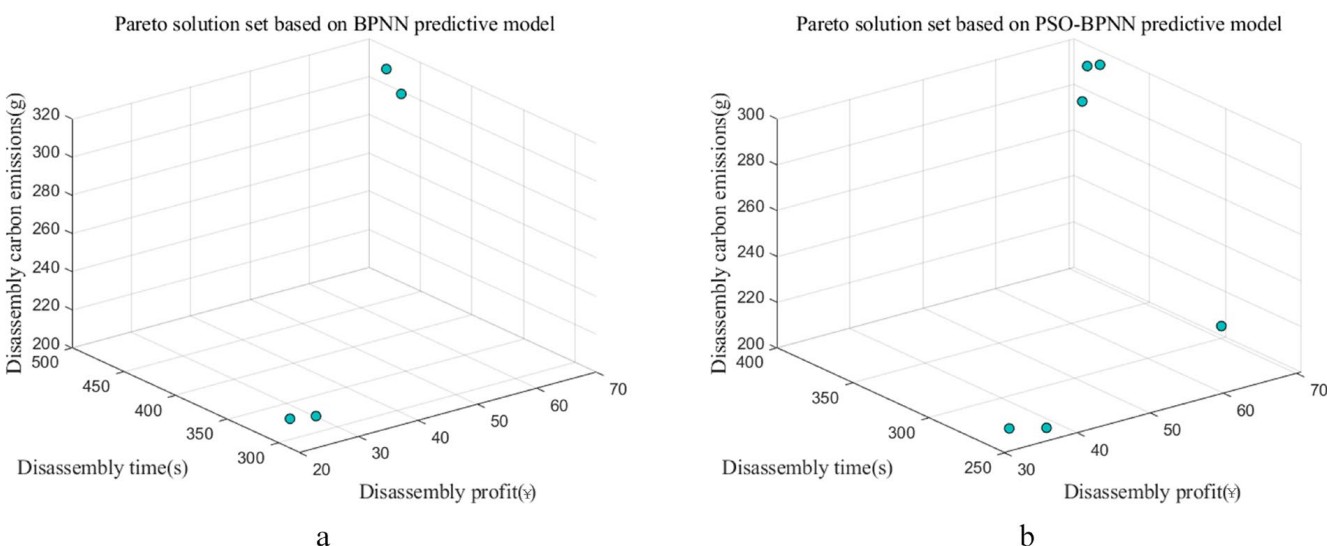

**Figure 13.** The spatial distribution of pareto solution ((**a**) BPNN; (**b**) PSO-BPNN).

The two pareto solution sets show that the overall quality of the solution and various optimization indictors are all different. Particularly, in terms of the disassembly time and disassembly carbon emissions, the method proposed in this study has shown significant improvements. Based on the optimization results shown in Table 6, compared to group A, the number of non-inferior solutions in group B increased by five, while the number

of optimized solutions increased by three, showing the superior overall solution quality. Although the top three disassembly depths in group A and group B are the same, the overall performance indicators in group B are significantly better than those in group A: a 16.8% reduction in disassembly time, 6.45% lower carbon emissions, and a 53.54% increase in disassembly profit compared to group A. Another interesting result is observed by analyzing the third disassembly depths of the two groups, which achieve the same disassembly profit in both groups; group B demonstrates a 17.1% shorter disassembly time and 7.8% lower carbon emissions compared to group A. The reason for the above results can be attributed to the unfeasibility of the optimization model, which failed to fully capture the target information, resulting in inaccurate predictions and causing errors in the subsequent disassembly depth optimization process.

**Table 6.** Comparison of disassembly depth optimization results.

| Group Number | No. | The Disassembly Depth | Disassembly Time (s) | Disassembly Profit (¥) | Disassembly Carbon Emissions (g) | The Non-Inferior Solution | The Optimized Solution |
|---|---|---|---|---|---|---|---|
| A | 1 | 1-2-3-5-6-7-8 | 279.5 | 22.72 | 218.6 | × | × |
| | 2 | 1-2-3-6-8-5 | 290.5 | 20.39 | 217.7 | × | × |
| | 3 | 1-2-3-6-10-5-8-7-11-12 | 471.5 | 70.58 | 315.6 | × | × |
| | 4 | 1-2-3-5-6-8-11-12 | 416.5 | 60.98 | 319.4 | √ | × |
| B | 1 | 1-2-3-5-6-7-8 | 221.5 | 63.19 | 207.8 | √ | √ |
| | 2 | 1-2-3-6-8-5 | 254.5 | 36.94 | 204.9 | √ | × |
| | 3 | 1-2-3-6-10-5-8-7-11-12 | 390.5 | 70.58 | 290.7 | √ | √ |
| | 4 | 1-2-3-6-7-5-8-9-11-12 | 382.5 | 70.31 | 293.8 | √ | √ |
| | 5 | 1-2-3-6-5-7-8-11-9-12 | 368.5 | 64.90 | 287.3 | √ | × |
| | 6 | 1-2-3-6-7-8 | 264.3 | 33.62 | 203.1 | √ | × |

To further verify the feasibility of the proposed method in this study, the disassembly time, disassembly profit, and disassembly carbon emissions were selected as optimization objectives and iterated 100 times. The results are shown in Figure 14, Figure 15 and Figure 16, respectively.

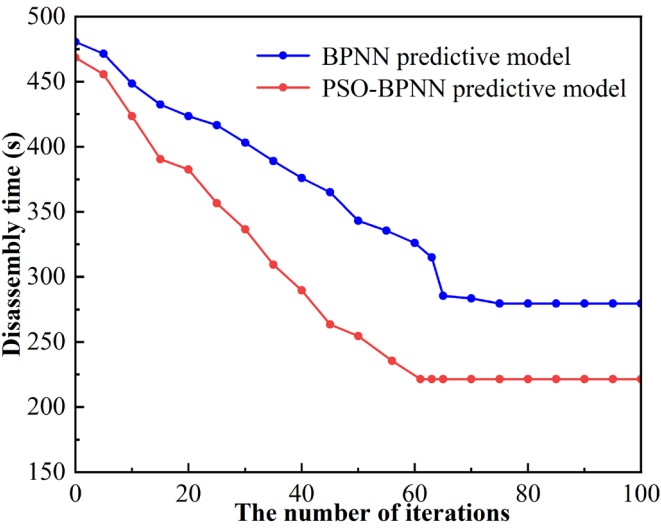

**Figure 14.** The iteration graph of disassembly time.

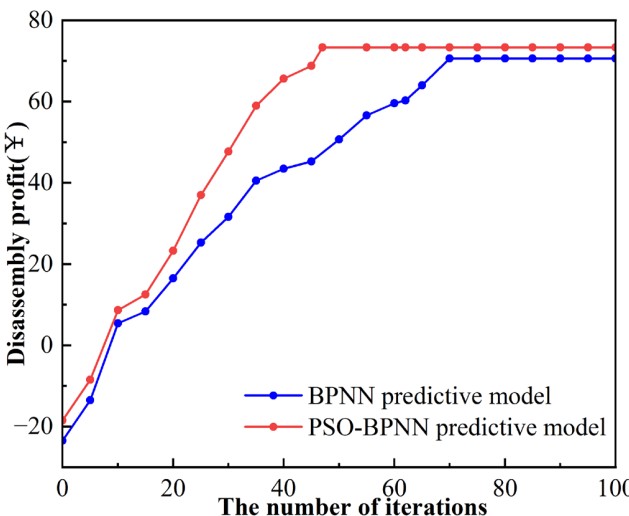

**Figure 15.** The iteration graph of disassembly profit.

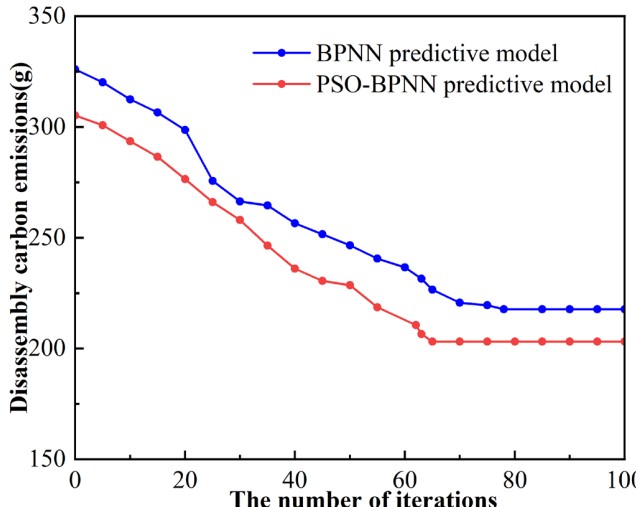

**Figure 16.** The iteration graph of disassembly carbon emissions.

During the process of searching for the optimized solutions of disassembly time, as shown in Figure 14, the PSO-BPNN predictive model tends to converge at the *61*-th generation, compared to the *75*-th generation for the BPNN predictive model, which represents an 18.6% improvement in convergence speed. Figure 15 shows, that in the process of searching for an optimized disassembly profit, the PSO-BPNN predictive model converges at the *47*-th generation, compared to the *70*-th generation for the BPNN predictive model, which represents a 32.8% improvement in convergence speed. Figure 16 shows that, in the process of searching for optimized disassembly carbon emissions, the PSO-BPNN predictive model converges at the *65*-th generation, compared to the *78*-th generation for the BPNN predictive model, which represents a 16.6% improvement in convergence speed.

### 4.3. The Limitations

The proposed method in this study provides a reference direction for the disassembly of EoL electronic products, combined with a multi-objective optimization algorithm and deep learning model. However, there are also some limitations. Firstly, the definition of the disassembly rules is based on the needs of most disassembly companies, which potentially limits their generalizability. Secondly, although the combination of BPNN and NSGA-II shows an effective optimization ability during the optimization process, there is still room

for improvement in terms of fitness function design, data quality, and algorithm efficiency. Finally, the dataset adequately represents the diversity of the disassembly process; the dataset size can be further expanded to enhance the generalization capability of the model.

## 5. Conclusions

In this study, an improved method based on the PSO-BPNN predictive model is proposed, which aims to shorten disassembly time, reduce disassembly carbon emissions, and increase disassembly profit. The PSO-BPNN model directly responds to the traditional neural network and mathematical model, which lacks accuracy in face of multi-objective nonlinear problems. Then, the feasibility of the proposed method is verified by the 'Huawei P7' disassembly case. Compared to the BPNN predictive model, the PSO-BPNN predictive model significantly reduced the MSE approximately 92.95% for disassembly time, 96.51% for disassembly profit, and 92.74% for disassembly carbon emissions, which demonstrates superior performance and provides a feasible optimization model for disassembly depth optimization. Based on the PSO-BPNN predictive model, the proposed method maintains disassembly profit while reducing disassembly time and disassembly carbon emissions by 17.1% and 7.8%, respectively. Additionally, the convergence speed of the PSO-BPNN predictive model's search for optimal solutions in disassembly time, disassembly profit, and disassembly carbon emissions increased by 18.6%, 32.8%, and 16.6%, respectively. In summary, the proposed method provides high-quality pareto solutions and offers a reference for research on the disassembly of EoL electronic products.

In future research, additional optimization objectives such as disassembly complexity and remanufacturing potential can be considered. In addition, other more superior network models can also be considered for optimization.

**Author Contributions:** Conceptualization, S.J., L.L., and Y.Y.; methodology, S.J. and L.L.; software, S.J.; validation, S.J.; formal analysis, L.L., F.Y., and Y.Y.; investigation, L.L., Y.Y., and F.Y.; resources, Y.Y. and F.Y.; data curation, L.L., Y.Y., and F.Y.; writing—original draft preparation, S.J.; writing—review and editing, L.L. and F.Y.; supervision, L.L. and F.Y.; project administration, L.L. and F.Y.; funding acquisition, L.L. All authors have read and agreed to the published version of the manuscript.

**Funding:** This work was supported by the National Key Research and Development Program of China (grant number 2020YFB1713001).

**Institutional Review Board Statement:** Not applicable.

**Informed Consent Statement:** Not applicable.

**Data Availability Statement:** The original contributions presented in this study are included in the article. Further inquiries can be directed to the corresponding author(s).

**Acknowledgments:** The authors wish to thank Huaqing Li for his guidance with the BPNN model and Xiaojing Chu for her help in collecting the disassembly experiment data.

**Conflicts of Interest:** The authors declare no conflicts of interest. The funders had no role in the design of the study; in the collection, analyses, or interpretation of data; in the writing of the manuscript; or in the decision to publish the results.

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
