# Peer review of "An Improved Method for Disassembly Depth Optimization of End-of-Life Smartphones Based on PSO-BP Neural Network Predictive Model"

_sustainability, doi:10.3390/su17209032_

Round 1

Reviewer 1 Report

Comments and Suggestions for Authors
  1. The introduction needs to define the concept of “disassembly depth optimization” more clearly. The authors frequently mention it, but the essence of this task only becomes clear in Section 2.
  2. The novelty of the research is poorly justified and should be elaborated on more deeply.
  3. How is the disassembly precedence graph (Figure 2) constructed? Is it built by a human expert or generated automatically by a program?
  4. The algorithm described on page 7 is a standard Particle Swarm Optimization applied to tune a neural network. This is a fairly standard task that does not present significant novelty.
  5. It is unclear how the dataset for training the neural network was assembled. Am I correct in understanding that specialists disassembled multiple Huawei P7 units in different sequences, each time measuring the duration? What is the size of this dataset?
  6. How does the constructed model predict carbon emissions? If this value is directly proportional to the time spent on disassembly, why is it necessary to predict it separately?

Reviewer 2 Report

Comments and Suggestions for Authors

Thank you for the submission. I believe this study has potential for publication. Please find my feedback as a comment in the uploaded PDF document.

Round 2

Reviewer 1 Report

Comments and Suggestions for Authors

My concered issues have been addressed. This paper can be accepted.

Reviewer 2 Report

Comments and Suggestions for Authors

The authors of the study have answered all the queries, and they have made all necessary revisions accordingly.